# Superconductivity of Co-Doped CaKFe_4_As_4_ Investigated via Point-Contact Spectroscopy and London Penetration Depth Measurements

**DOI:** 10.3390/nano14151319

**Published:** 2024-08-05

**Authors:** Erik Piatti, Daniele Torsello, Francesca Breccia, Tsuyoshi Tamegai, Gianluca Ghigo, Dario Daghero

**Affiliations:** 1Department of Applied Science and Technology, Politecnico di Torino, I-10129 Torino, Italy; daniele.torsello@polito.it (D.T.); gianluca.ghigo@polito.it (G.G.); dario.daghero@polito.it (D.D.); 2Istituto Nazionale di Fisica Nucleare, Sezione di Torino, I-10125 Torino, Italy; 3Department of Applied Physics, The University of Tokyo, Bunkyo-ku, Tokyo 113-8656, Japan

**Keywords:** iron-based superconductors, layered superconductors, multiband superconductivity, point-contact Andreev reflection spectroscopy, London penetration depth

## Abstract

The iron-based superconductors (IBSs) of the recently discovered 1144 class, unlike many other IBSs, display superconductivity in their stoichiometric form and are intrinsically hole doped. The effects of chemical substitutions with electron donors are thus particularly interesting to investigate. Here, we study the effect of Co substitution in the Fe site of ${\mathrm{CaKFe}}_{4}$${\mathrm{As}}_{4}$ single crystals on the critical temperature, on the energy gaps, and on the superfluid density by using transport, point-contact Andreev-reflection spectroscopy (PCARS), and London penetration depth measurements. The pristine compound ($T_{c}≃36$ K) shows two isotropic gaps whose amplitudes ($\Delta_{1}$ = 1.4–3.9 meV and $\Delta_{2}$ = 5.2–8.5 meV) are perfectly compatible with those reported in the literature. Upon Co doping (up to ≈7% Co), $T_{c}$ decreases down to ≃20 K, the spin-vortex-crystal order appears, and the low-temperature superfluid density is gradually suppressed. PCARS and London penetration depth measurements perfectly agree in demonstrating that the nodeless multigap structure is robust upon Co doping, while the gap amplitudes decrease as a function of $T_{c}$ in a linear way with almost constant values of the gap ratios $2\Delta_{i}/k_{B}T_{c}$.

## 1. Introduction

Iron-based superconductors (IBSs) are an extensive group of different families of layered materials which are built up by stacking basic building blocks of two-dimensional iron–pnictogen or iron–chalcogen layers [1,2]. These two-dimensional iron-based layers are pivotal in supporting complex phase diagrams, tunable by both doping and pressure [3,4,5], where unconventional superconductivity with critical temperatures up to 56 K [6,7] is intertwined with complex magnetic ordering [8,9,10], making these compounds extremely appealing from both fundamental and applied perspectives [2,11,12]. In recent years, interest has focused on IBS families composed by the intergrowth of multiple basic building blocks, such as the generalized 122 structure where superconducting ${\mathrm{Fe}}_{2}$${\mathrm{As}}_{2}$ layers are alternated by spacer layers. For example, the intergrowth of 122-type *A*${\mathrm{Fe}}_{2}$${\mathrm{As}}_{2}$ (*A* being an alkali metal) and 1111-type CaFeAsF layers results in the highly anisotropic 12442 fluoroarsenide family [13,14], which exhibits striking resemblances to double-layer cuprates [15,16,17] down to the possibly nodal character of their superconducting order parameter [18,19,20,21,22].

Conversely, the intergrowth of two 122-type layers with alternating Ae (alkali-earth) and *A* atoms results in the 1144 (AeA${\mathrm{Fe}}_{4}$${\mathrm{As}}_{4}$) family [23,24], which originally attracted significant interest owing to its stoichiometric nature supporting superconducting critical temperatures up to $T_{c}\approx 36$ K in the absence of doping or pressure. This made the 1144 family very attractive for investigating the fundamental properties of the superconducting state in IBSs, since it provided a platform where undesired effects introduced by dopant atoms could be avoided. Indeed, a nodeless $s_{\pm }$ symmetry of the superconducting order parameter—with a strong multiband character, placed firmly in the clean limit and aided by spin fluctuations—was found to be compatible with both experimental and theoretical findings [25,26,27,28,29,30,31,32,33,34,35].

Like in most IBSs [4,36], electron doping can be induced in 1144 compounds via the aliovalent substitution of a transition metal (usually Co or Ni) at the Fe site, leading to the emergence of complex and rich phase diagrams [37,38]. For instance, in ${\mathrm{RbEuFe}}_{4}$${\mathrm{As}}_{4}$—where the parent compound exhibits $T_{c}\approx 36$ K and ferromagnetic ordering below 15 K [24]—Ni substitution at the Fe site tunes the material from a superconducting ferromagnet to a ferromagnetic superconductor and promotes the emergence of a spin-density wave phase [37]. Conversely, in ${\mathrm{CaKFe}}_{4}$${\mathrm{As}}_{4}$, the parent compound is superconducting at $T_{c}\approx 35$ K and exhibits magnetic fluctuations [31,39] but no static magnetic ordering [40]. Upon either Ni or Co substitution at the Fe site [38], this compound develops a peculiar antiferromagnetic (AFM) state known as spin-vortex crystal (SVC) with the so-called hedgehog structure [38,41,42]. Contrary to most IBSs, this AFM state emerges without nematic ordering and is associated with the two non-equivalent As sites introduced by the alternated Ca and K spacer layers in the 1144 structure [43].

The superconducting gap structure of the ${\mathrm{CaKFe}}_{4}$${\mathrm{As}}_{4}$ parent compound has been extensively investigated by means of angle-resolved photoemission spectroscopy (ARPES) [27], scanning tunneling spectroscopy (STS) [25,30], nuclear magnetic resonance [31], London penetration depth ($\lambda_{L}$) measurements [25,29,33,44], and break-junction spectroscopy [35]—all pointing to the existence of at least two effective nodeless gaps $\Delta_{1}\in \left[6,10\right]$ meV and $\Delta_{2}\in \left[1,4\right]$ meV. To date, investigations on how electron doping and the associated emergence of SVC magnetic ordering affect the superconducting gap structure of ${\mathrm{CaKFe}}_{4}$${\mathrm{As}}_{4}$ have instead been more sparse and limited to the Ni-doped compound [26,42,45]—whereas the Co-doped compound remains mostly unexplored—and the doping dependence of the gap amplitudes is still missing.

Here, we report on a study of the evolution of the critical temperature, of the energy gaps and of the superfluid density as a function of the doping content *x* in CaK(${\mathrm{Fe}}_{1-x}$,${\mathrm{Co}}_{x}$)_4_${\mathrm{As}}_{4}$ single crystals from x=0 to x≈0.07, by means of transport measurements, point-contact Andreev reflection spectroscopy (PCARS) [46,47,48,49] and coplanar-waveguide resonator (CPWR) measurements [50,51] of the London penetration depth $\lambda_{L}$. Through this exhaustive set of measurements, we assess the evolution of multigap superconductivity in this compound with increasing Co doping. We find the amplitudes of the superconducting gaps to be isotropic with no evidence of nodal lines at any doping level, and to be well described by an effective two-band *s*-wave model in both PCARS and CPWR measurements. Upon increasing Co doping, the gap amplitudes decrease while maintaining nearly constant gap-to-critical temperature ratios with a slight increase at the largest doping level x≈0.07, which is in line with the typical behavior displayed by other IBSs. These results confirm that the $s_{\pm }$ gap structure is robust against Co doping and suggests that the SVC magnetic ordering does not significantly affect the superconducting pairing in this compound.

## 2. Materials and Methods

### 2.1. Crystal Growth

Single crystals of CaK(${\mathrm{Fe}}_{1-x}$,${\mathrm{Co}}_{x}$)_4_${\mathrm{As}}_{4}$ were grown by the self-flux method using FeAs as described in Ref. [52] following the protocol reported in Refs. [53,54]. The chemical compositions of all crystals were analyzed by using energy-dispersive X-ray spectroscopy. Analyzed Co-doping levels in the crystals [x=0.023(4), 0.046(5), and 0.073(2)] are lower than the corresponding nominal compositions (x=0.03, 0.07, and 0.09, respectively). The analyzed levels are reported throughout the manuscript with the uncertainty omitted for brevity.

### 2.2. Resistivity Measurements

The electrical resistivity of the crystals was measured via the van der Pauw method as described in Ref. [22]. The temperature was controlled by loading the samples either on the cold finger of a ST-403 pulse-tube cryocooler (Cryomech, Syracuse, NY, USA) or on a home-built cryogenic insert which was directly immersed in the He vapors in a He storage dewar.

### 2.3. Point-Contact Andreev Reflection Spectroscopy Measurements

PCARS measurements were performed by measuring the current–voltage (I−V) characteristic of point-like contacts between a normal-metal wire and the superconducting crystals in the pseudo-four-probe configuration. The point contacts were realized via the soft technique [46,47,48], and the direction of the (main) current injection was controlled by placing them on the sides (ab-plane injection) of the regular platelet-like crystals, as detailed in Refs. [19,49,55]. Differential conductance (dI/dV) spectra were obtained by numerical derivation of the I−V characteristics, which were normalized to the normal-state spectrum measured immediately above $T_{c}$ as customary for IBSs [46,49,56]. To account for the spreading resistance due to the non-zero resistivity of the crystal in the region of the resistive transition, the normalization was accomplished by vertically shifting the normal-state conductance curve and by rescaling the voltage as described in Refs. [36,57]. The normalized curves were fitted to the two-band version [46,47] of the isotropic two-dimensional Blonder–Tinkham–Klapwijk model [58,59]. The temperature was controlled using the same cryogenic insert employed in the resistivity measurements.

### 2.4. Superfluid Density Measurements

The superfluid density $\varrho_{s}$ of each sample was measured by means of a coplanar waveguide resonator (CPWR) technique, which is particularly suitable to study small crystals with a critical temperature in the range of 20–60 K [60]. The measurement device consists of an ${\mathrm{YBa}}_{2}$${\mathrm{Cu}}_{3}$$O_{7-\delta }$ film patterned as a coplanar waveguide resonator, to which the sample is coupled. The measurement is carried out within a resonator perturbation approach that gives access to the absolute value of the London penetration depth $\lambda_{L}$ and its temperature dependence by measuring resonant frequency shifts and variations of the unloaded quality factor as a function of temperature and performing a calibration procedure [51]. The temperature-dependent experimental superfluid density is then obtained as $\varrho_{s}\left(T\right)=\lambda_{L}{\left(T\right)}^{-2}$. The CPWR measurements also allow extracting the temperature-dependent surface impedance ($Z_{s}=R_{s}+iX_{s}$) of the sample as described in Refs. [61,62].

## 3. Results

We first assess the evolution of the electric transport properties of our CaK(${\mathrm{Fe}}_{1-x}$,${\mathrm{Co}}_{x}$)_4_${\mathrm{As}}_{4}$ single crystals upon increasing Co content. Figure 1a shows the temperature dependence of the in-plane electrical resistivity ρ(T) of four crystals at different Co doping levels (x=0, 0.023, 0.046, and 0.073). The room-temperature resistivity is ρ(300K)≈350μΩ cm in the undoped crystal, which is in good agreement with the literature [26,40], and it monotonically increases with increasing Co doping up to ρ(300K)≈415μΩ cm in the crystal doped with *x* = 0.073.

On decreasing the temperature *T*, all resistivity curves show a convex *T*-dependence with a smooth change in slope around 200 K, which is usually associated with an incoherent–coherent crossover in IBSs where the dominant charge carriers are holes [15,26]. Below ∼100 K, the curvature changes to concave, and then ρ(T) drops to zero as superconductivity develops in the system. The resistivity immediately above the resistive transition, $\rho^{\mathrm{on}}$, increases from ≈21 to ≈88μΩ cm from x=0 to x=0.073, corresponding to a decrease in the residual resistivity ratio from ≈16 to ≈4.7, which is in good agreement with previous results [26,38] and as expected due to the increase in disorder introduced by Co substitution.

As shown in Figure 1b, in the undoped crystal, the resistive transition to the superconducting state occurs at $T_{c}^{\rho }=35.8$ K (midpoint) and is extremely sharp ($\Delta T_{c}^{\rho }<0.1$ K), reflecting the high quality of the present samples. The introduction of Co dopants in the system progressively reduces the critical temperature and broadens the transition, reaching $T_{c}^{\rho }=21.3$ K and $\Delta T_{c}^{\rho }\approx 1.0$ K in the crystal with x=0.073, which is consistent with the increase in disorder detected in the normal state.

While the normal-state resistivity of the undoped crystal remains smooth in the temperature range between $T_{c}^{\rho }$ and 100 K, an additional feature can be observed in the ρ(T) of the Co-doped samples at sufficiently large Co doping, where the slope suddenly changes [38]. While not being direct evidence for the occurrence of the magnetic transition, this kink has been shown to map well the Néel temperature $T_{N}$ where the SVC magnetic order develops in the system [26,38], and it can be more accurately tracked by considering the *T* dependence of the first derivative of the resistivity, dρ/dT [26,38]. As shown in Figure 1c, the slope change in ρ(T) translates to a step-like change in dρ/dT, which can be clearly distinguished from the curved background due to the closing of the SC transition in the crystals at x=0.046 and 0.073. Following Ref. [38], we estimate $T_{N}$ as the foot of the step-like change, which we determine here as the intersection between the extrapolations of the linear fits of the dρ/dT data immediately above and below it. This protocol allows us to estimate the magnetic ordering to develop at a maximum $T_{N}=39.6\pm 2.2$ K in the crystal at x=0.073, which decreases as the Co doping is reduced, as expected [26,38].

Figure 1d summarizes the phase diagram of our CaK(${\mathrm{Fe}}_{1-x}$,${\mathrm{Co}}_{x}$)_4_${\mathrm{As}}_{4}$ single crystals as mapped by the electric transport measurements. We see that our crystals exhibit the same behavior displayed by Co-doped ${\mathrm{CaKFe}}_{4}$${\mathrm{As}}_{4}$ in the literature [38]: namely, Co doping progressively weakens superconductivity and suppresses $T_{c}^{\rho }$ while simultaneously leading to the emergence of the resistive anomaly associated to SVC magnetic order and increasing its onset temperature, $T_{N}$. Interestingly, however, in the present study, $T_{N}$ becomes detectable as the step-like change in dρ/dT as early as x=0.046, while in Ref. [38], it could not be clearly distinguished until x=0.07.

The superconducting gap structure in our CaK(${\mathrm{Fe}}_{1-x}$,${\mathrm{Co}}_{x}$)_4_${\mathrm{As}}_{4}$ single crystals was primarily assessed by means of PCARS measurements, which exploit the Andreev reflection at the interface between the metallic and superconducting sides of a point-contact junction. This is a quantum mechanical phenomenon where an electron impinging on the interface from the metallic side with an energy smaller than the energy gap Δ of the superconductor is reflected back as a hole, enabling the transmission of a Cooper pair into the superconductor [58,63]. This enhances (up to a factor of 2) the differential conductance of the point-contact junction at bias voltages smaller than Δ/e and embeds it with fundamental information on the superconducting energy gap and its properties in the direct and reciprocal space (for a more detailed and quantitative description, we refer the reader to specific reviews [46,47,48]). In particular, the differential conductance spectra of multigap superconductors exhibit peaks or shoulders which are the hallmarks of the different gaps $\Delta_{i}$ [46,47,48].

The PCARS measurements discussed here were technically difficult because of the very small thickness of the crystals (of the order of 10 μm). We successfully obtained conductance curves with spectroscopic signals only by cutting or breaking the crystals and then making the contact on the freshly exposed side. This means that we always injected the current along the ab planes and thus in a different configuration with respect to the STS measurements in ${\mathrm{CaKFe}}_{4}$${\mathrm{As}}_{4}$ reported in the literature [25,30] that were made with the current injected along the *c* axis. In our CaK(${\mathrm{Fe}}_{1-x}$,${\mathrm{Co}}_{x}$)_4_${\mathrm{As}}_{4}$ single crystals, all PCAR spectra clearly show multiple gap features, which was as expected due to the multiband nature of the compound. In no cases was it possible to fit the experimental spectra by using a single-gap model. Even though ARPES measurements [27] indicate the existence of four different gaps, we systematically used a model with two isotropic gaps in order to keep the number of adjustable parameters to a minimum.

Figure 2a,b show two examples of unnormalized (as-measured) conductance curves as a function of *T* in crystals with x=0.023 (a) and x=0.073 (b). The enhancement of the conductance at low bias is due to Andreev reflection; its amplitude decreases on increasing *T* and disappears when the curves recorded at different *T* start to overlap. This is the critical temperature of the contact, $T_{c}^{A}$, which usually falls between the onset and the completion of the resistive transition measured by transport. Experimentally, due to the finite *T* step between different spectra, we defined $T_{c}^{A}$ as the midpoint between the temperature of the last superconducting spectrum and the first normal-state one. The relevant uncertainty $\Delta T_{c}^{A}$ is thus one-half of the *T* step between the two curves. In many cases, the low-*T* spectra display typical “dips” at high bias, which are due to the attainment of the critical current in the contact region [36,46,64,65]. These features will not be taken into account here, as they are not included in the models for Andreev reflection, unless one adds a specific current-dependent Maxwell-like term in the contact resistance [36].

The resistance of the contacts in the normal state, $R_{n}$, inferred from the high-bias value of the differential conductance curves at low *T*, is indicated in the labels. When $R_{n}$ is not very large, as in Figure 2a, the spectra display a downward shift (accompanied by a horizontal stretching) in correspondence with the transition of the bulk to the resistive state. These effects are due to the spreading resistance [36,57,66,67] of the sample itself—which is not negligible due to the small thickness of the crystals—and impose some caution in the normalization. In particular, one has to remove the contribution of the (current-dependent) spreading resistance both from the differential conductance and from the voltage scale in order to find the correct “normal-state conductance” $G_{n}\left(V\right)$ [36,67]. The spreading resistance being actually unknown, there is some degree of uncertainty in the determination of $G_{n}\left(V\right)$. To account for such uncertainty, we always tried different normalizations and fitted the resulting normalized spectra, thus obtaining a range of gap values for each contact.

As an example, Figure 2c shows a symmetrized and normalized low-temperature spectrum (hollow blue circles) for each of the Co contents. Clearly, each of these spectra corresponds to a specific point contact and was obtained by dividing the raw spectrum by a specific choice of the relevant $G_{n}\left(V\right)$. The fit with the two-band *s*-wave model is represented by a solid red line. The agreement between the fitting function and the experimental data is impressive at low bias in the energy range of the gaps; however, some additional structures at higher energy can sometimes appear, i.e., the aforementioned dips or shoulders that are not included in the fitting function. The amplitudes of the gaps extracted from the fit are indicated in the labels, and their decrease upon increasing the Co content is evident; in particular, for x=0.073, they are approximately one-half of those in the undoped crystals.

The superconducting gap structure was also independently probed by determining the *T* dependence of the superfluid density $\varrho_{s}$ via CPWR measurements. Figure 3a shows the experimental evolution of $\varrho_{s}$ with *T* (symbols) for four crystals at the different values of x=0, 0.023, 0.046, and 0.073. It can be seen that increasing the Co concentration suppresses $\varrho_{s}$ in the entire *T* range and the temperature at which it goes to zero, i.e., the superconducting critical temperature $T_{c}^{\lambda }$. As shown in Figure 3b (violet diamonds, left scale), $T_{c}^{\lambda }$ decreases from 35.7 K at x=0 to 21.1 K at x=0.073, which correlates very well with the values of $T_{c}^{\rho }$ determined from the transport measurements. At the same doping levels, $\varrho_{s}\left(7\;K\right)$ is suppressed from 32 to 6.8 μ$m^{-2}$. The experimental $\varrho_{s}\left(T\right)$ data from all doping levels of our samples series can then be fitted with a BCS model involving two *s*-wave gaps [68,69,70]:(1)$$\varrho_{s}\left(T\right)=\frac{1}{\lambda_{L}{\left(0\right)}^{2}}\sum_{i}w_{i}\left[1+\frac{1}{\pi }\int_{0}^{2\pi }\int_{\Delta_{i}\left(T\right)}^{\infty }\frac{\partial f}{\partial E}\frac{EdEd\phi }{\sqrt{E^{2}-\Delta_{i}^{2}\left(T\right)}}\right]$$
where *i* identifies the band, $\Delta_{i}\left(T\right)$ is the *s*-wave superconducting gap function, $f={\left[1+\exp\left(E/k_{B}T\right)\right]}^{-1}$ is the Fermi function and $w_{i}$ is the mixing weight of the *i*-th gap contribution (constrained by $w_{1}+w_{2}=1$). The resulting fitting functions are displayed in Figure 3a as solid lines and closely follow the experimental data, as evidenced by the small relative deviation displayed in Figure 3c. This agreement fully supports the multiband *s*-wave symmetry and gives gap values perfectly consistent with those determined by PCARS, as discussed more extensively in the next section. In addition, the fits give also access to the zero-temperature value of the penetration depth, $\lambda_{L}\left(0\right)$. As shown in Figure 3b (green circles, right scale), the aforementioned suppression of $\varrho_{s}$ results in an increase in the value of $\lambda_{L}\left(0\right)$ from 172 nm at x=0 to 373 nm at x=0.073, which can be ascribed to enhanced carrier scattering [45].

This enhanced carrier scattering introduced by Co doping is also confirmed by considering the *T* dependence of the surface impedance $Z_{s}$, which is displayed in Figure 3c. Beyond the gradual increase in the normal-state resistance with increasing Co content, all samples show a peak in the imaginary part of the surface impedance, $X_{s}$, just below $T_{c}^{\lambda }$. The existence of this peak can be explained within a two-fluid model, where it arises because upon entering the superconducting state, the initial reduction in normal electrons is not immediately compensated, in terms of screening of the microwave field, by the increase of the superconducting current [61]. The temperature at which the peak is found depends linearly on the measurement frequency (about 8 GHz in our case) and on the carrier relaxation time, and it becomes therefore suppressed when carrier scattering is increased by the increasing Co doping, resulting in an increased spread between the peak temperature and $T_{c}$.

## 4. Discussion

The overall dependence of the gaps on the doping content is shown in Figure 4b, which includes the gap values obtained both from PCARS (filled symbols) and CPWR (hollow symbols) measurements. In the former, the error bars on the gap amplitudes indicate the uncertainty arising from the normalization of the PCARS spectra, and each point represents the average gap amplitude in a specific contact. It is clear that both $\Delta_{1}$ and $\Delta_{2}$, even though spread in a certain range, decrease almost linearly on increasing *x*, as highlighted by the shaded regions. Both in the undoped compound and for all the Co contents explored here, the energy ranges of the two gaps never overlap and are instead rather well separated. In particular, in the undoped compound, the amplitudes of the gaps obtained in different point contacts are $\Delta_{1}\in \left[1.4,3.9\right]$ meV and $\Delta_{2}\in \left[5.2,8.5\right]$ meV if the error bars are included. These ranges of values agree rather well with the continuous distribution with two probability peaks observed by STM [25,30] and reported in Figure 4a, even though our PCARS measurements were made with the current injected along the ab planes and not along the *c* axis as in STM. They also agree with the results of incoherent multiple Andreev-reflection effect spectroscopy in planar break-junctions [35], superfluid density measurements [29,44] and London penetration depth measurements [25]. Why none of these techniques give results in agreement with ARPES [27] remains a puzzle to be solved. As a matter of fact, ARPES spectra gave evidence of three large gaps ranging between 11 and 13 meV and of a fourth gap of about 8 meV, with no indications about any gap smaller than that [27]. Incidentally, these gaps did not allow explaining either the field dependence of the coherence length or the temperature dependence of the superfluid density as obtained from muon spin rotation (μSR) [71] that instead require the existence of an additional smaller gap of about 2.4 meV.

The intensity of the signal associated to one gap in PCARS is, roughly speaking, dependent on the cross-sectional area of the corresponding Fermi surface and should thus not change much from contact to contact [46,47,48]. In the great majority of the PCARS spectra we obtained, the fit required assigning a dominant weight (between 70% and 80%) to the smaller gap. It is worth mentioning that the Fermi surface sheet with the largest cross-section is the outer holelike one, labeled with γ [27,72] which, according to ARPES, hosts the gap of about 8 meV. This is another discrepancy with respect to ARPES results that deserves further investigation. Interestingly, the PCARS weights turn out to be universal for all doping contents. Since Co doping should affect in opposite ways electronic and holonic Fermi surface sheets [27,73], this robustness might indicate that the gaps we are measuring actually reside in Fermi surface sheets of the same kind, i.e., the hole surfaces—even though we have no direct way to associate the gaps to the Fermi surface pockets.

Concerning the weights used in the fits to the CPWR measurements, one must first note that the weights that appear in the fitting functions for the superfluid density [68,69,70] and for the normalized PCARS spectra [46,47,48] are based on different properties of the Fermi surfaces and therefore do not necessarily coincide. For our CPWR analysis, we initially adopted the value $w_{1}=0.37$ for the weight of the smaller gap. This was the value calculated in Ref. [71] for the $\lambda_{L}^{-2}$ analysis based on muon spin rotation measurements. This choice turns out to work well for all the Co-doped crystals but not for the pristine one. Here, the data are instead best described by a predominant contribution of the smaller gap ($w_{1}=0.67$), which is close to the weights used in PCARS. Despite the discrepancy in the used weights, our analysis of the $\lambda_{L}^{-2}\left(T\right)$ data gives values of the smaller gap (2.5 meV) and of $\lambda_{L}\left(0\right)$ (172 nm) for the pristine crystal, which are very close to the results reported in Ref. [71] (2.4 meV and 187 nm, respectively).

More interesting information is provided by Figure 4c, which shows the gap amplitudes measured by PCARS in different contacts as a function of the Andreev critical temperature $T_{c}^{A}$ of the contacts themselves (filled symbols) and those measured by CPWR as a function of $T_{c}^{\lambda }$ (hollow symbols). The first thing one can notice is the extremely good agreement between the two data sets with the gaps extracted from CPWR measurements sitting almost perfectly in the center of the cloud of data from PCARS. Unlike in Figure 4b, the shaded regions of Figure 4c have a specific physical meaning since they are bound by lines of equation $\Delta_{i}=c_{i}k_{B}T_{c}/2$ where $c_{i}$ is a constant. In other words, these lines correspond to constant gap ratios $c_{i}=2\Delta_{i}/k_{B}T_{c}$, that are $c_{1}$ = 1.0–2.7 for the small gap $\Delta_{1}$ and $c_{2}$ = 3.6–5.7 for the large gap $\Delta_{2}$. The solid red line corresponds to the ideal BCS gap ratio $2\Delta /k_{B}T_{c}=3.53$ for a single-band *s*-wave superconductor, to highlight that, as in ${\mathrm{MgB}}_{2}$ and in most of the two-band superconductors, one gap is smaller and the other is larger than the BCS value [74,75].

In the most doped crystals, the values of $\Delta_{2}$ gather on the upper part of the shaded region or even slightly above it. This fact indicates an apparent increase in the large gap ratio when the critical temperature is of the order of 20 K, which is in perfect agreement with a general observation based on the comparison of various measurements in different families of IBSs [48,76]. The increase in the gap ratio can be explained by recalling that superconductivity in these compounds is mediated by electronic excitations (spin fluctuations) and that the temperature evolution of the spin-resonance energy [77] follows that of the superconducting energy gap. This suggests the existence of a feedback effect of the condensate on the spin fluctuations, that—once taken into account within the Eliashberg theory—allows explaining the increase in the gap ratio at low critical temperatures [48,78]. In this context, our measurements indicate that the interaction between superconductivity and the SVC ordering peculiar to the electron-doped ${\mathrm{CaKFe}}_{4}$${\mathrm{As}}_{4}$ compounds does not qualitatively differ from the one observed in other IBSs characterized by less exotic forms of AFM ordering.

Finally, an important contribution toward a comprehensive understanding of the role of doping in this compound should come from the analysis of the statistical properties of disorder introduced by Co substitution. Specifically, it would be interesting to investigate whether long-range power-law correlations exist and how they impact on the superconducting properties, as recently discussed in several papers [79,80,81]. However, this clearly goes beyond the scope of this paper, since our characterization techniques do not allow such study, but we hope our results will stimulate further experiments in this direction.

## 5. Conclusions

The stoichiometric 1144 compounds, and in particular ${\mathrm{CaKFe}}_{4}$${\mathrm{As}}_{4}$, have been extensively characterized and studied in the literature. The presence of multiple bands crossing the Fermi level [32,72,73,82] is known to give rise to a multigap structure [25,27,29,30,33,35,83] that can be understood theoretically as an example of the $s_{\pm }$ symmetry with a coupling mechanism mediated by spin fluctuations [27,31]. These systems are intrinsically hole doped, and heterovalent substitutions with higher-valence elements in the Fe site progressively suppress the superconducting properties, leading to the appearance of a peculiar magnetic order known as spin-vortex crystal [40,43]. The $s_{\pm }$ gap structure has been shown to be robust against Ni doping, but a determination of the doping dependence of the gap amplitudes is still missing.

In this paper, we have combined transport measurements, point-contact Andreev reflection spectroscopy and London penetration depth measurements to study the evolution of the critical temperature, of the energy gaps and of the superfluid density as a function of the doping content *x* in CaK(${\mathrm{Fe}}_{1-x}$,${\mathrm{Co}}_{x}$)_4_${\mathrm{As}}_{4}$ single crystals, up to x=0.073. The superfluid density as a function of temperature, as well as the Andreev-reflection spectra, were fitted by using a two-band model. In the pristine compound, the gap amplitudes $\Delta_{1}=\left[1.4,3.9\right]$ meV and $\Delta_{2}=\left[5.2,8.5\right]$ meV were found to agree very well with previous findings in the literature [25,27,29,35]. In the Co-doped crystals, the gap structure remains the same, with two well distinct isotropic gaps whose amplitudes decrease linearly with increasing *x*. The gap ratios $2\Delta_{i}/k_{B}T_{c}$ remain almost constant, within the experimental uncertainty due to the spread of gap values, and slightly increase just in the most doped crystals, which is probably due to the feedback effect of the condensate on the spin fluctuations that mediate superconductivity [78]. The relative weights of the gaps determined from the point-contact spectra remain approximately constant upon doping, which may indicate that all the measured gaps pertain to Fermi surface sheets of the same kind, i.e., hole surfaces. Moreover, the linearity of the gap trends (similar to that observed in other IBSs of different families, e.g., Fe(Te,Se) [84], Ba(Fe,Co)_2_${\mathrm{As}}_{2}$ [85], ${\mathrm{RbCa}}_{2}$${\mathrm{Fe}}_{4}$${\mathrm{As}}_{4}$$F_{2}$ [19]) suggest that the development of the magnetic order (here of the hedgehog spin-vortex crystal type) does not affect superconductivity more than in other IBSs characterized by more conventional antiferromagnetic orders.

## Figures and Tables

**Figure 1 nanomaterials-14-01319-f001:**
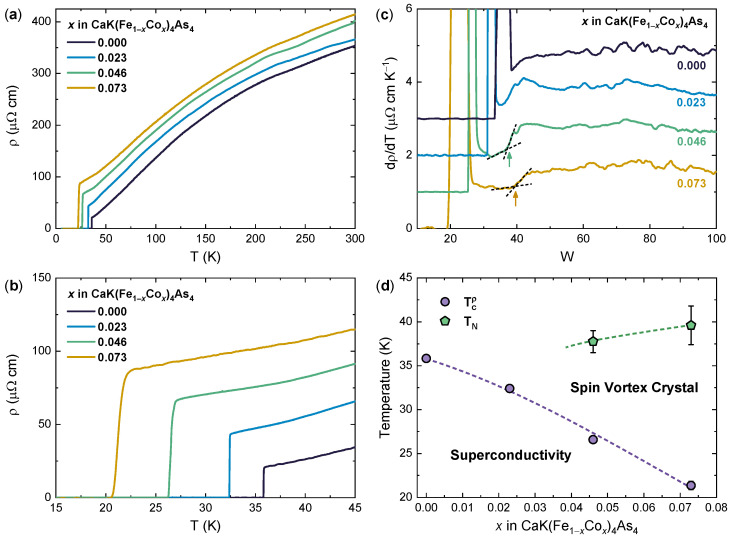
Electric transport in CaK(${\mathrm{Fe}}_{1-x}$,${\mathrm{Co}}_{x}$)_4_${\mathrm{As}}_{4}$ single crystals. (**a**) In-plane resistivity ρ as a function of temperature *T* for an undoped sample (x=0) and three samples with increasing Co doping (x=0, 0.023, 0.046, and 0.073). (**b**) Low-*T* magnification of (**a**) which highlights the onset of superconductivity. The transition temperature $T_{c}^{\rho }$ is defined as the midpoint of each resistive transition. (**c**) *T* dependence of dρ/dT obtained by numerical derivation of the curves shown in (**a**). Arrows highlight the step-like increase associated with SVC magnetic ordering at the Néel temperature $T_{N}$, as estimated by the intersection of the linear fits above and below $T_{N}$ (dashed black lines). The curves are vertically offset for clarity. (**d**) *T*–*x* phase diagram determined from transport measurements. Data points for $T_{c}^{\rho }$ and $T_{N}$ as a function of *x* mark the boundaries of the two main ordered phases, superconductivity and SVC magnetic order. Dashed lines are guides to the eye.

**Figure 2 nanomaterials-14-01319-f002:**
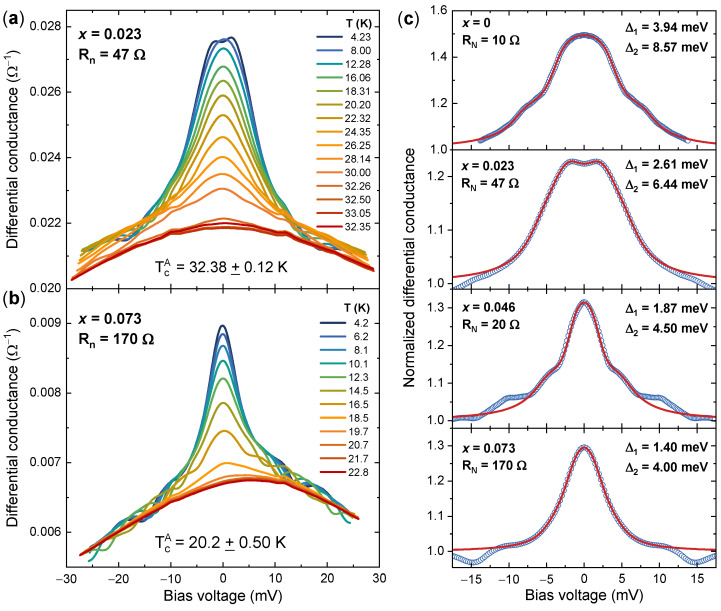
Point-contact Andreev-reflection measurements in CaK(${\mathrm{Fe}}_{1-x}$,${\mathrm{Co}}_{x}$)_4_${\mathrm{As}}_{4}$ single crystals. (**a**) Differential conductance dI/dV as a function of the bias voltage in an ab-plane point-contact on a crystal with x=0.023. The normal-state resistance of the contact and its critical temperature are indicated in the label. (**b**) Same as in (**a**) but for an ab-plane point contact on a crystal with x=0.073. (**c**) Examples of normalized spectra at low temperature (hollow blue circles) and the relevant fit with the two-band model (solid red lines) for the different Co contents. All the spectra were taken at 4.2 K. The normal-state resistance and the amplitudes of the gaps are indicated in the labels.

**Figure 3 nanomaterials-14-01319-f003:**
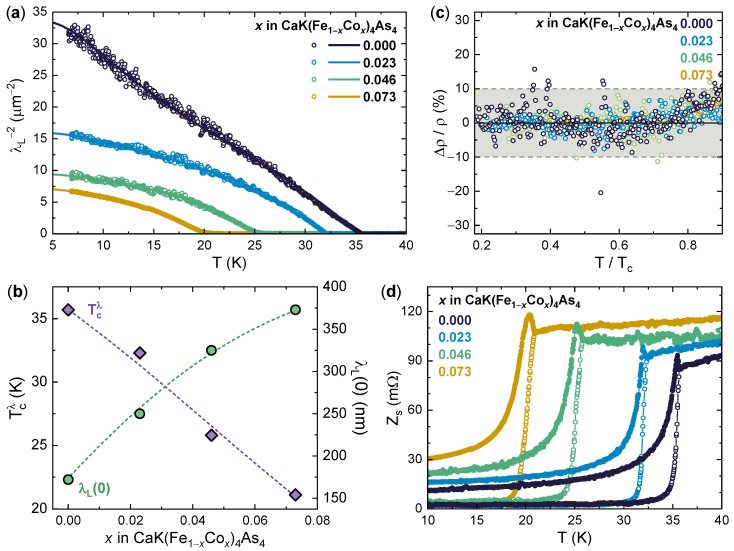
Coplanar waveguide resonator measurements in CaK(${\mathrm{Fe}}_{1-x}$,${\mathrm{Co}}_{x}$)_4_${\mathrm{As}}_{4}$ single crystals. (**a**) Temperature dependence of the superfluid density $\varrho_{s}\left(T\right)=\lambda_{L}^{-2}\left(T\right)$ measured by CPWR (symbols) on CaK(${\mathrm{Fe}}_{1-x}$,${\mathrm{Co}}_{x}$)_4_${\mathrm{As}}_{4}$ single crystals for an undoped sample (x=0) and three samples with increasing Co doping (x=0.023, 0.046, and 0.073). Solid lines are the BCS fits to the experimental data according to Equation (Equation 1). (**b**) Superconducting critical temperature $T_{c}^{\lambda }$ (violet diamonds, left scale) determined from the CPWR measurements and $\lambda_{L}\left(0\right)$ obtained from the $\varrho_{s}\left(T\right)$ fits (green circles, right scale) as a function of *x*. Dashed lines are guides to the eye. (**c**) Relative deviation between the experimental data and the fits in panel (**a**). The shadowed area indicates the ±10% region. (**d**) Temperature dependence of the surface impedance $Z_{s}$, separated in its real part $R_{s}$ (hollow circles) and imaginary part $X_{s}$ (filled circles), as measured by CPWR. Solid lines are guides to the eye.

**Figure 4 nanomaterials-14-01319-f004:**
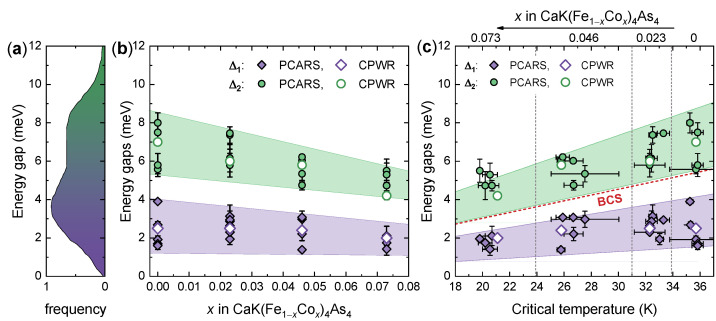
Doping and critical temperature dependence of the superconducting gaps in CaK(${\mathrm{Fe}}_{1-x}$,${\mathrm{Co}}_{x}$)_4_${\mathrm{As}}_{4}$ single crystals. (**a**) An example of the distribution of gap amplitudes obtained by STM in the pristine compound (taken from Ref. [25]), which is in very good agreement with the results obtained by PCARS and CPWR in crystals with x=0. (**b**) Gap amplitudes $\Delta_{1}$ and $\Delta_{2}$ as a function of the doping content *x*. Error bars on the PCARS data indicate the range of gap values obtained by fitting the same curve with different normalizations, and symbols indicate the midpoint of this range. Colored regions are just guides to the eye and highlight a linear-with-doping suppression in both the gap amplitudes. (**c**) Energy gaps as a function of the critical temperature of the contacts (PCARS) or of the critical temperature where the superfluid density goes to zero (CPWR). The meaning of vertical error bars is the same as in (**b**), while horizontal error bars indicate the uncertainty on the critical temperature. Colored regions are bounded by lines corresponding to constant gap ratios $2\Delta_{i}/k_{B}T_{c}$. The dashed red line corresponds to the BCS gap ratio $2\Delta /k_{B}T_{c}=3.53$. The Co content to which the points refer is indicated on top of the panel and increases from right to left.

## Data Availability

All data needed to evaluate the conclusions in the paper are included in the manuscript. Additional data related to this paper may be obtained from the corresponding author upon reasonable request.

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
