# Peer review of "Superconductivity of Co-Doped CaKFe4As4 Investigated via Point-Contact Spectroscopy and London Penetration Depth Measurements"

_nanomaterials, 2024, doi:10.3390/nano14151319_

Round 1

Reviewer 1 Report

Comments and Suggestions for Authors

This is a good piece of experimental work where Piatti et al. discuss the study of the superconducting gap structure in Co-doped 1144 sample series by means of point-contact Andreev reflection spectroscopy and London penetration depth measurements. The obtained superfluid density as a function of temperature, as well as the Andreev-reflection spectra, were fitted using a two-band model allowing for the determination of the absolute values of the big and small superconducting gaps. The gaps were found to scale with T_c. 

The paper is well written. The amount and quality of the experimental data support all the main conclusions made in the paper. I recommend the paper for publication after considering the following points:

1. I have some doubts about the determination of the magnetic transition based on resistivity data (Figs. 1c and 1d). The step-like change of dρ/dT is seen for the 0.023 sample, while it starts to wash away for the x=0.0073 one. Even the undoped sample shows the beginning of the anomaly. How do the authors decide that the 0.046 and 0.073 samples have a magnetic transition and the 0.000 and 0.0023 ones do not?

2. The anomalies in Fig. 1c look different from those reported in Ref. 39. In Ref. 39, the anomaly in dR(T)/dT is very sharp, while here I see nothing comparable with the Ref. 39 data. I am not sure, therefore, that the SVC points in Fig. 1d represent the magnetic transition. Anyway, the "phase diagram" in Fig. 1d is not important for further discussion and interpretation of PCAR and \lambda^-2 data.

3. The higher gap value for the undoped sample (x=0, Fig. 2c) is smaller compared to that obtained in ARPES experiments. Following [PRL 117 277001 (2016)], the average value of the larger gap should be around 10 to 12 meV.

4. The deviation of the experimental \lambda^-2(T) points from the data is quite substantial (Fig. 3a). In particular, the blue points (x=0.0023 sample) stay systematically above the fitted curve for temperatures ranging from 10 to 25 K, i.e., more than in half of the 'superconducting' temperature range. The points for the x=0.00 sample deviate from the fit below T~10 K. I suggest the author present the "deviation curves", i.e., differences between the data and the fits. An explanation for the deviation for x=0.00 and x=0.0023 is also needed.

5. The contribution of the larger and the smaller gaps to the superfluid density and PCAR data were estimated to be ~25\% and 75\%. This looks a bit strange since the large superconducting gaps were found to open at least in 4 Fermi surface sheets [see PRL 117 277001 (2016)], while the smaller gaps were not detected by means of ARPES. In addition, the muon experiments [PRB 97 140503 (2018)] report the contribution of the larger gap as ~70\%, which seems to better correspond to the ARPES data. I would also mention that PCAR experiments were performed in a different geometry compared to that of the penetration depth studies. The gap contributions should not remain the same for different geometries. 

Reviewer 2 Report

Comments and Suggestions for Authors

The authors experimentally studied the superconducting critical
temperature, the superconducting gap, and superfluid density in Co-doped
IBS as a function of doping. The topic and the results are interesting. The
manuscript is well formulated. Therefore, in my opinion the manuscript
deserves to be published.

Reviewer 3 Report

Comments and Suggestions for Authors

Review analysis for manuscript nanomaterials-3106646 - entitled “Superconductivity of Co-doped CaKFe4As4 investigated via point-contact spectroscopy and London penetration depth measurements”, authored by Erik Piatti, Daniele Torsello, Francesca Breccia, Tsuyoshi Tamegai, Gianluca Ghigo and Dario Daghero

The proposed manuscript studies the doping effects of Co/Fe substitution on the superconducting system CaKFe(Co)4As4 in bulk (single crystal) form. The investigation is focused on several key parameters: Tc, resistive properties, penetration depth and electronic nature: energy gaps and superfluid density. The manuscript treats a challenging open question for exploring the relation between the doping and superconducting state. This is achieved by implementation of advance experimental techniques as point contact Andreev-reflection spectroscopy and coplanar-waveguide resonator measurements. The analysis and the discussion are comprehensive and very detailed. The manuscript can be accepted with a minor revision. I would like to propose to authors few suggestions for possible further improvement of the discussion.

Please, specify the explicit basic physics relation of the
point-contact Andreev-reflection technique and the differential conductance for electronic structure analysis, especially for the determination of the band gaps. Please, add the fitting function of the two-band s-wave model used for evaluation of the band gaps.

Please, note additionally the chemical quality (phase purity) of the investigated crystals. The Co/Fe substitution percentage is low enough, presumably below the solubility limit. Following this context, the presence of other chemical inclusions can be excluded (as eventual carrier scattering centres for instance). Is there any indirect assessment for altered carrier concentration with the Co-doping or the superconducting order parameter is mainly affected by the intrinsic structural disorder/magnetic order caused by the substitution?

Reviewer 4 Report

Comments and Suggestions for Authors

Referee Report on the Manuscript titled "Superconductivity of Co-doped CaKFe4As4 investigated via point-contact spectroscopy and London penetration depth measurements" by Erik Piatti et al.

In this manuscript, the authors investigate the effects of cobalt (Co) substitution at the iron (Fe) site in CaKFe4As4 single crystals. The study focuses on critical temperature (Tc), energy gaps, and superfluid density, employing transport measurements, point-contact Andreev-reflection spectroscopy (PCARS), and London penetration depth measurements. The results are analyzed using a two-band model. The authors reveal that the gap structure remains unchanged with Co doping, characterized by two distinct isotropic gaps whose amplitudes decrease linearly with increasing Co concentration. The gap ratios 2∆i/kBTc ​ remain nearly constant within experimental uncertainty. Upon Co doping up to approximately 7%, the critical temperature decreases to about 20 K, spin-vortex-crystal order emerges, and the low-temperature superfluid density is gradually suppressed. These findings suggest that the nodeless multigap structure is robust under Co doping, although the gap amplitudes decrease linearly with Tc, with nearly constant gap ratios.

The subject matter of this paper is scientifically interesting and pertinent to ongoing research in the field of superconductivity. The findings are accurate and contribute significantly to advancing the understanding of Co-doped CaKFe4As4, justifying publication in Nanomaterials. The manuscript is well-structured and effectively summarizes the main findings.

Before publication, I recommend that the authors address the following points:

-        Recently, the role of long-range power law disorder correlations in superconducting characteristics has gained significant interest. Studies have demonstrated that such correlations can influence the statistical properties of Cooper pair correlations and the superconducting order parameter, enhancing superconducting correlations with increased disorder correlation degree.{Neverov, V.D. et al., "Correlated disorder as a way towards robust superconductivity." Commun. Phys. 2022, 5, 177.} Similar conclusions have been made for the temperature dependent superconducting characteristics. { de Braganca, R. et al., "Impact of Correlated Disorder on Surface Superconductivity: Revealing the Robustness of the Surface Ordering Effect," J. Phys. Chem. Lett. 2024, 15, 2573} These studies suggest that correlated disorder can significantly impact superconducting properties. Researchers led by A. Bianconi identified that in lanthanum copper oxide, the arrangement of oxygen ions in interstitial layers follows a power law distribution, which is crucial for the exceptional conduction characteristics of cuprates. {Fratini, M. et al., "Scale-free structural organization of oxygen interstitials in La2CuO4+y." Nature 2010, 466, 841}

In light of this, I request that the authors discuss the potential role of the statistical properties of the disorder introduced by Co substitution in their compounds. Specifically, can the Co-induced disorder exhibit similar long-range correlations, and could it play a role in the superconducting properties observed? The authors should provide a discussion on this aspect within the article.

Addressing this issue will enhance the manuscript's depth and provide a more comprehensive understanding of the influence of disorder on superconductivity in Co-doped CaKFe4As4.

The manuscript presents significant and accurate findings that advance the knowledge of Co-doped CaKFe4As4 superconductors. However, a discussion on the role of disorder, particularly in the context of long-range power law correlations, would further enrich the study. I recommend the manuscript for publication in Nanomaterials following the authors' incorporation of the requested discussion.
